# A Block-Coordinate Descent Approach for Large-scale Sparse Inverse Covariance Estimation

**Eran Treister**[*][†]
Computer Science, Technion, Israel
and Earth and Ocean Sciences, UBC
Vancouver, BC, V6T 1Z2, Canada
eran@cs.technion.ac.il

**Javier Turek**[*]
Department of Computer Science
Technion, Israel Institute of Technology
Technion City, Haifa 32000, Israel
javiert@cs.technion.ac.il

## Abstract

The sparse inverse covariance estimation problem arises in many statistical applications in machine learning and signal processing. In this problem, the inverse of a covariance matrix of a multivariate normal distribution is estimated, assuming that it is sparse. An $\ell_1$ regularized log-determinant optimization problem is typically solved to approximate such matrices. Because of memory limitations, most existing algorithms are unable to handle large scale instances of this problem. In this paper we present a new block-coordinate descent approach for solving the problem for large-scale data sets. Our method treats the sought matrix block-by-block using quadratic approximations, and we show that this approach has advantages over existing methods in several aspects. Numerical experiments on both synthetic and real gene expression data demonstrate that our approach outperforms the existing state of the art methods, especially for large-scale problems.

## 1 Introduction

The multivariate Gaussian (Normal) distribution is ubiquitous in statistical applications in machine learning, signal processing, computational biology, and others. Usually, normally distributed random vectors are denoted by $\mathbf{x} \sim \mathcal{N}(\mu, \mathbf{\Sigma}) \in \mathbb{R}^{\mathbf{n}}$, where $\mu \in \mathbb{R}^{\mathbf{n}}$ is the mean, and $\mathbf{\Sigma} \in \mathbb{R}^{\mathbf{n} \times \mathbf{n}}$ is the covariance matrix. Given a set of realizations $\{\mathbf{x}_i\}_{i=1}^m$, many such applications require estimating the mean $\mu$, and either the covariance $\mathbf{\Sigma}$ or its inverse $\mathbf{\Sigma}^{-1}$, which is also called the precision matrix. Estimating the inverse of the covariance matrix is useful in many applications [2] as it represents the underlying graph of a Gaussian Markov Random Field (GMRF). Given the samples $\{\mathbf{x}_i\}_{i=1}^m$, both the mean vector $\mu$ and the covariance matrix $\mathbf{\Sigma}$ are often approximated using the standard maximum likelihood estimator (MLE), which leads to $\hat{\mu} = \frac{1}{m} \sum_{i=0}^m \mathbf{x}_i$ and[1]

$$\mathbf{S} \stackrel{\triangle}{=} \hat{\mathbf{\Sigma}}^{\mathrm{MLE}} = \frac{1}{m} \sum_{i=0}^m (\mathbf{x}_i - \hat{\mu})(\mathbf{x}_i - \hat{\mu})^T, \tag{1}$$

which is also called the empirical covariance matrix. Specifically, according to the MLE, $\mathbf{\Sigma}^{-1}$ is estimated by solving the optimization problem

$$\min_{\mathbf{A} \succ 0} f(\mathbf{A}) \stackrel{\triangle}{=} \min_{\mathbf{A} \succ 0} -\log(\det \mathbf{A}) + \mathrm{tr}(\mathbf{S}\mathbf{A}), \tag{2}$$

---

[*]The authors contributed equally to this work.

[†]Eran Treister is grateful to the Azrieli Foundation for the award of an Azrieli Fellowship.

[1]Equation (1) is the standard MLE estimator. However, sometimes the unbiased MLE estimation is preferred, where $m - 1$ replaces $m$ in the denominator.

which is obtained by applying $-\log$ to the probability density function of the Normal distribution. However, if the number of samples is lower than the dimension of the vectors, i.e., $m < n$, then $\mathbf{S}$ in (1) is rank deficient and not invertible, whereas the true $\mathbf{\Sigma}$ is assumed to be positive definite, hence full-rank. Still, when $m < n$ one can estimate the matrix by adding further assumptions. It is well-known [5] that if $(\mathbf{\Sigma}^{-1})_{ij} = 0$ then the random scalar variables in the $i$-th and $j$-th entries in $\mathbf{x}$ are conditionally independent. Therefore, in this work we adopt the notion of estimating the *inverse* of the covariance, $\mathbf{\Sigma}^{-1}$, assuming that it is sparse. (Note that in most cases $\mathbf{\Sigma}$ is dense.) For this purpose, we follow [2, 3, 4], and minimize (2) with a sparsity-promoting $\ell_1$ prior:

$$\min_{\mathbf{A} \succ 0} F(\mathbf{A}) \triangleq \min_{\mathbf{A} \succ 0} f(\mathbf{A}) + \lambda \|\mathbf{A}\|_1. \tag{3}$$

Here, $f(\mathbf{A})$ is the MLE functional defined in (2), $\|\mathbf{A}\|_1 \equiv \sum_{i,j} |a_{ij}|$, and $\lambda > 0$ is a regularization parameter that balances between the sparsity of the solution and the fidelity to the data. The sparsity assumption corresponds to a small number of statistical dependencies between the variables. Problem (3) is also called *Covariance Selection* [5], and is non-smooth and convex.

Many methods were recently developed for solving (3)—see [3, 4, 7, 8, 10, 11, 12, 15, 16] and references therein. The current state-of-the-art methods, [10, 11, 12, 16], involve a "proximal Newton" approach [20], where a quadratic approximation is applied on the smooth part $f(\mathbf{A})$ in (3), leaving the non-smooth $\ell_1$ term intact, in order to obtain the Newton descent direction. To obtain this, the gradient and Hessian of $f(\mathbf{A})$ are needed and are given by

$$\nabla f(\mathbf{A}) = \mathbf{S} - \mathbf{A}^{-1}, \qquad \nabla^2 f(\mathbf{A}) = \mathbf{A}^{-1} \otimes \mathbf{A}^{-1}, \tag{4}$$

where $\otimes$ is the Kronecker product. The gradient in (4) already shows the main difficulty in solving this problem: it contains $\mathbf{A}^{-1}$, the inverse of the sparse matrix $\mathbf{A}$, which may be dense and expensive to compute. The advantage of the proximal Newton approach for this problem is the low overhead: by calculating the $\mathbf{A}^{-1}$ in $\nabla f(\mathbf{A})$, we also get the Hessian at the same cost [11, 12, 16].

In this work we aim at solving large scale instances of (3), where $n$ is large, such that $O(n^2)$ variables cannot fit in memory. Such problem sizes are required in fMRI [11] and gene expression analysis [9] applications, for example. Large values of $n$ introduce limitations: (a) They preclude storing the full matrix $\mathbf{S}$ in (1), and allow us to use only the vectors $\{\mathbf{x}_i\}_{i=1}^m$, which are assumed to fit in memory. (b) While the sparse matrix $\mathbf{A}$ in (3) fits in memory, its dense inverse does not. Because of this limitation, most of the methods mentioned above cannot be used to solve (3), as they require computing the full gradient of $f(\mathbf{A})$, which is a dense $n \times n$ symmetric matrix. The same applies for the blocking strategies of [2, 7], which target the *dense* covariance matrix itself rather than its inverse, using the dual formulation of (3). One exception is the proximal Newton approach in [11], which was made suitable for large-scale matrices by treating the Newton direction problem in blocks.

In this paper, we introduce an iterative Block-Coordinate Descent [20] method for solving large-scale instances of (3). We treat the problem in blocks defined as subsets of columns of $\mathbf{A}$. Each block sub-problem is solved by a quadratic approximation, resulting in a descent direction that corresponds only to the variables in the block. Since we consider one sub-problem at a time, we can fully store the gradient and Hessian for the block. In contrast, [11] applies a blocking approach to the full Newton problem, which results in a sparse $n \times n$ descent direction. There, all the columns of $\mathbf{A}^{-1}$ are calculated for the gradient and Hessian of the problem for each *inner* iteration when solving the full Newton problem. Therefore, our method requires less calculations of $\mathbf{A}^{-1}$ than [11], which is the most computationally expensive task in both algorithms. Furthermore, our blocking strategy allows an efficient linesearch procedure, while [11] requires computing a determinant of a sparse $n \times n$ matrix. Although our method is of linear order of convergence, it converges in less iterations than [11] in our experiments. Note that the asymptotic convergence of [11] is quadratic only if the *exact* Newton direction is found at each iteration, which is very costly for large-scale problems.

## 1.1 Newton's Method for Covariance Selection

The proximal Newton approach mentioned earlier is iterative, and at each iteration $k$, the smooth part of the objective in (3) is approximated by a second order Taylor expansion around the $k$-th iterate $\mathbf{A}^{(k)}$. Then, the Newton direction $\boldsymbol{\Delta}^*$ is the solution of an $\ell_1$ penalized quadratic minimization problem,

$$\min_{\boldsymbol{\Delta}} \ \tilde{F}(\mathbf{A}^{(k)} + \boldsymbol{\Delta}) = \min_{\boldsymbol{\Delta}} f(\mathbf{A}^{(k)}) + \mathrm{tr}(\boldsymbol{\Delta}(\mathbf{S} - \mathbf{W})) + \frac{1}{2}\mathrm{tr}(\boldsymbol{\Delta}\mathbf{W}\boldsymbol{\Delta}\mathbf{W}) + \lambda\|\mathbf{A}^{(k)} + \boldsymbol{\Delta}\|_1, \quad (5)$$

where $\mathbf{W} = (\mathbf{A}^{(k)})^{-1}$ is the inverse of the $k$-th iterate. Note that the gradient and Hessian of $f(\mathbf{A})$ in (4) are featured in the second and third terms in (5), respectively, while the first term of (5) is constant and can be ignored. Problem (5) corresponds to the well-known LASSO problem [18], which is popular in machine learning and signal/image processing applications [6]. The methods of [12, 16, 11] apply known LASSO-solvers for treating the Newton direction minimization (5).

Once the direction $\boldsymbol{\Delta}^*$ is computed, it is added to $\mathbf{A}^{(k)}$ employing a linesearch procedure to sufficiently reduce the objective in (3) while ensuring positive definiteness. To this end, the updated iterate is $\mathbf{A}^{(k+1)} = \mathbf{A}^{(k)} + \alpha^*\boldsymbol{\Delta}^*$, and the parameter $\alpha^*$ is obtained using Armijo's rule [1, 12]. That is, we choose an initial value of $\alpha_0$, and a step size $0 < \beta < 1$, and accordingly define $\alpha_i = \beta^i \alpha_0$. We then look for the smallest $i \in \mathbb{N}$ that satisfies the constraint $\mathbf{A}^{(k)} + \alpha_i\boldsymbol{\Delta}^* \succ 0$, and the condition

$$F(\mathbf{A}^{(k)} + \alpha_i\boldsymbol{\Delta}^*) \le F(\mathbf{A}^{(k)}) + \alpha_i\sigma \left[\mathrm{tr}(\boldsymbol{\Delta}^*(\mathbf{S} - \mathbf{W})) + \lambda\|\mathbf{A}^{(k)} + \boldsymbol{\Delta}^*\|_1 - \lambda\|\mathbf{A}^{(k)}\|_1\right]. \quad (6)$$

The parameters $\alpha_0$, $\beta$, and $\sigma$ are usually chosen as $1, 0.5$, and $10^{-4}$ respectively.

## 1.2 Restricting the Updates to Active Sets

An additional significant idea of [12] is to restrict the minimization of (5) at each iteration to an "active set" of variables and keep the rest as zeros. The active set of a matrix $\mathbf{A}$ is defined as

$$\mathbf{Active}(\mathbf{A}) = \left\{(i,j) : \mathbf{A}_{ij} \ne 0 \vee |(\mathbf{S} - \mathbf{A}^{-1})_{ij}| > \lambda\right\}. \quad (7)$$

This set comes from the definition of the sub-gradient of (3). In particular, as $\mathbf{A}^{(k)}$ approaches the solution $\mathbf{A}^*$, $\mathbf{Active}(\mathbf{A}^{(k)})$ approaches $\{(i,j) : \mathbf{A}^*_{ij} \ne 0\}$. As noted in [12, 16], restricting (5) to the variables in $\mathbf{Active}(\mathbf{A}^{(k)})$ reduces the computational complexity: given the matrix $\mathbf{W}$, the Hessian (third) term in (5) can be calculated in $O(Kn)$ operations instead of $O(n^3)$, where $K = |\mathbf{Active}(\mathbf{A}^{(k)})|$. Hence, any method for solving the LASSO problem can be utilized to solve (5) effectively while saving computations by restricting its solution to $\mathbf{Active}(\mathbf{A}^{(k)})$. Our experiments have verified that restricting the minimization of (5) only to $\mathbf{Active}(\mathbf{A}^{(k)})$ does not significantly increase the number of iterations needed for convergence.

## 2 Block-Coordinate-Descent for Inverse Covariance (BCD-IC) Estimation

In this Section we describe our contribution. To solve problem (3), we apply an iterative Block-Coordinate-Descent approach [20]. At each iteration, we divide the column set $\{1, ..., n\}$ into blocks. Then we iterate over all blocks, and in turn minimize (3) restricted to the "active" variables of each block, which are determined according to (7). The other matrix entries remain fixed during each update. The matrix $\mathbf{A}$ is updated after each block-minimization.

We choose our blocks as sets of columns because the portion of the gradient (4) that corresponds to such blocks can be computed as solutions of linear systems. Because the matrix is symmetric, the corresponding rows are updated simultaneously. Figure 1 shows an example of a BCD iteration where the blocks of columns are chosen in sequential order. In practice, the sets of columns can be non-contiguous and vary between the BCD iterations. We elaborate later on how to partition

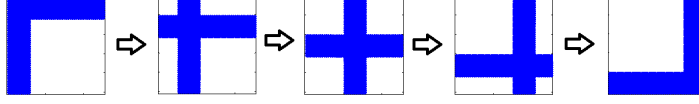

Figure 1: Example of a BCD iteration. The blocks are treated successively.

the columns, and on some advantages of this block-partitioning. Partitioning the matrix into small blocks enables our method to solve (3) in high dimensions (up to millions of variables), requiring $O(n^2/p)$ additional memory, where $p$ is the number of blocks (that is in addition to the memory needed for storing the iterated solution $\mathbf{A}^{(k)}$ itself).

## 2.1 Block Coordinate Descent Iteration

Assume that the set of columns $\{1, ..., n\}$ is divided into $p$ blocks $\{I_j\}_{j=1}^p$, where $I_j$ is the set of indices that corresponds to the columns and rows in the $j$-th block. As mentioned before, in the BCD-IC algorithm we traverse all blocks and update the iterated solution matrix block by block. We denote the updated matrix after treating the $j$-th block at iteration $k$ by $\mathbf{A}_j^{(k)}$ and the next iterate $\mathbf{A}^{(k+1)}$ is defined once the last block is treated, i.e., $\mathbf{A}^{(k+1)} = \mathbf{A}_p^{(k)}$.

To treat each block of (3), we adopt both of the ideas described earlier: we use a quadratic approximation to solve each block, while also restricting the updated entries to the active set. For simplicity of notation in this section, let us denote the updated matrix $\mathbf{A}_{j-1}^{(k)}$, before treating block $j$ at iteration $k$, by $\tilde{\mathbf{A}}$. To update block $j$, we change only the entries in the rows/columns in $I_j$. First, we form and minimize a quadratic approximation of problem (3), restricted to the rows/columns in $I_j$:

$$\min_{\boldsymbol{\Delta}_j} \ \tilde{F}(\tilde{\mathbf{A}} + \boldsymbol{\Delta}_j), \tag{8}$$

where $\tilde{F}(\cdot)$ is the quadratic approximation of (3) around $\tilde{\mathbf{A}}$, similarly to (5), and $\boldsymbol{\Delta}_j$ has non-zero entries only in the rows/columns in $I_j$. In addition, the non-zeros of $\boldsymbol{\Delta}_j$ are restricted to $\mathbf{Active}(\tilde{\mathbf{A}})$ defined in (7). That is, we restrict the minimization (8) to

$$\mathbf{Active}_{I_j}(\tilde{\mathbf{A}}) = \mathbf{Active}(\tilde{\mathbf{A}}) \cap \{(i,k) : i \in I_j \vee k \in I_j\}, \tag{9}$$

while all other elements are set to zero for the entire treatment of the $j$-th block. To calculate this set, we check the condition in (7) only in the columns and rows of $I_j$. To define this active set, and to calculate the gradient (4) for block $I_j$, we first calculate the columns $I_j$ of $\tilde{\mathbf{A}}^{-1}$, which is the main computational task of our algorithm. To achieve that, we solve $|I_j|$ linear systems, with the canonical vectors $\mathbf{e}_l$ as right-hand-sides for each $l \in I_j$, i.e., $(\tilde{\mathbf{A}}^{-1})_{I_j} = \tilde{\mathbf{A}}^{-1}\mathbf{E}_{I_j}$. The solution of these linear systems can be achieved in various ways. Direct methods may be applied using the Cholesky factorization, which requires up to $O(n^3)$ operations. For large dimensions, iterative methods such as Conjugate Gradients (CG) are usually preferred, because the cost of each iteration is proportional to the number of non-zeros in the sparse matrix. See Section A.4 in the Appendix for details about the computational cost of this part of the algorithm.

### 2.1.1 Treating a Block-subproblem by Newton's Method

To get the Newton direction for the $j$-th block, we solve the LASSO problem (8), for which there are many available solvers [22]. We choose the Polak-Ribiere non-linear Conjugate Gradients (NLCG) method of [19] which, together with a diagonal preconditioner, was used to solve this problem in [22, 19]. We describe the NLCG algorithm in Apendix A.1. To use this method, we need to calculate the objective of (8) and its gradient efficiently.

The calculation of the objective in (8) is much simpler than the full version in (5), because only blocks of rows/columns are considered. Denoting $\mathbf{W} = \tilde{\mathbf{A}}^{-1}$, to compute the objective in (8) and its gradient we need to calculate the matrices $\mathbf{W}\boldsymbol{\Delta}_j\mathbf{W}$ and $\mathbf{S} - \mathbf{W}$ only at the entries where $\boldsymbol{\Delta}_j$ is

non-zero (in the rows/columns in $I_j$). These matrices are symmetric, and hence, only their columns are necessary. This idea applies for the $\ell_1$ term of the objective in (8) as well.

In each iteration of the NLCG method, the main computational task involves calculating $\mathbf{W}\boldsymbol{\Delta}_j\mathbf{W}$ in the columns of $I_j$. For that, we reuse the $I_j$ columns of $\tilde{\mathbf{A}}^{-1}$ calculated for obtaining (9), which we denote by $\mathbf{W}_{I_j}$. Since we only need the result in the columns $I_j$, we first notice that $\left(\mathbf{W}\boldsymbol{\Delta}_j\mathbf{W}\right)_{I_j} = \mathbf{W}\boldsymbol{\Delta}_j\mathbf{W}_{I_j}$, and the product $\boldsymbol{\Delta}_j\mathbf{W}_{I_j}$ can be computed efficiently because $\boldsymbol{\Delta}_j$ is sparse.

Computing $\mathbf{W}(\boldsymbol{\Delta}_j\mathbf{W}_{I_j})$ is another relatively expensive part of our algorithm, and here we exploit the restriction to the Active Set. That is, we only need to compute the entries in (9). For this, we follow the idea of [11] and use the rows (or columns) of $\mathbf{W}$ that are represented in (9). Besides the columns $I_j$ of $\mathbf{W}$ we also need the "neighborhood" of $I_j$ defined as

$$N_j = \left\{i : \exists k \notin I_j : (i,k) \in \mathbf{Active}_{I_j}(\mathbf{A})\right\}. \tag{10}$$

The size of this set will determine the amount of additional columns of $\mathbf{W}$ that we need, and therefore we want it to be as small as possible. To achieve that, we define the blocks $\{I_j\}$ using clustering methods, following [11]. We use METIS [13], but other methods may be used instead. The aim of these methods is to partition the indices of the matrix columns/rows into disjoint subsets of relatively small size, such that there are as few as possible non-zero entries outside the diagonal blocks of the matrix that correspond to each subset. In our notation, we aim that the size of $N_j$ will be as small as possible for every block $I_j$, and that the size of $I_j$ will be small enough. Note that after we compute $\mathbf{W}_{N_j}$, we need to actually store and use only $|N_j| \times |N_j|$ numbers out of $\mathbf{W}_{N_j}$. However, there might be situations where the matrix has a few dense columns, resulting in some sets $N_j$ of size $O(n)$. Computing $\mathbf{W}_{N_j}$ for those sets is not possible because of memory limitations. We treat this case separately—see Section A.2 in the Appendix for details. For a discussion about the computational cost of this part—see Section A.4 in the Appendix.

### 2.1.2  Optimizing the Solution in the Newton Direction with Line-search

Assume that $\boldsymbol{\Delta}_j^*$ is the Newton direction obtained by solving problem (8). Now we seek to update the iterated matrix $\mathbf{A}_j^{(k)} = \mathbf{A}_{j-1}^{(k)} + \alpha^*\boldsymbol{\Delta}_j^*$, where $\alpha^* > 0$ is obtained by a linesearch procedure similarly to Equation (6).

For a general Newton direction matrix $\boldsymbol{\Delta}^*$ as in (6), this procedure requires calculating the determinant of an $n \times n$ matrix. In [11], this is done by solving $n-1$ linear systems of decreasing sizes from $n-1$ to 1. However, since our direction $\boldsymbol{\Delta}_j^*$ has a special block structure, we obtain a significantly cheaper linesearch procedure compared to [11], assuming that the blocks $I_j$ are relatively small. First, the trace and $\ell_1$ terms that are involved in the objective of (3) can be calculated with respect only to the entries in the columns $I_j$ (the rows are taken into account by symmetry). The $\log \det$ term, however, needs more special care, and is eventually reduced to calculating the determinant of an $|I_j| \times |I_j|$ matrix, which becomes cheaper as the block size decreases. Let us introduce a partitioning of any matrix $\mathbf{A}$ into blocks, according to a set of indices $I_j \subseteq \{1,...,n\}$. Assume without loss of generality that the rows and columns of $\mathbf{A}$ have been permuted such that the columns/rows with indices in $I_j$ appear first, and let

$$\mathbf{A} = \left[\begin{array}{c|c} \mathbf{A}_{11} & \mathbf{A}_{12} \\ \hline \mathbf{A}_{21} & \mathbf{A}_{22} \end{array}\right] \tag{11}$$

be a partitioning of $\mathbf{A}$ into four blocks. The sub-matrix $\mathbf{A}_{11}$ corresponds to the elements in rows $I_j$ and in columns $I_j$ in $\tilde{\mathbf{A}}$. According to the Schur complement [17], for any invertible matrix and block-partitioning as above, the following holds:

$$\log\det(\mathbf{A}) = \log\det(\mathbf{A}_{22}) + \log\det(\mathbf{A}_{11} - \mathbf{A}_{12}\mathbf{A}_{22}^{-1}\mathbf{A}_{21}). \tag{12}$$

In addition, for any symmetric matrix $\mathbf{A}$ the following applies:

$$\mathbf{A} \succ 0 \Leftrightarrow \mathbf{A}_{22} \succ 0 \text{ and } \mathbf{A}_{11} - \mathbf{A}_{12}\mathbf{A}_{22}^{-1}\mathbf{A}_{21} \succ 0. \tag{13}$$

Using the above notation for $\tilde{\mathbf{A}}$ and the corresponding partitioning for $\boldsymbol{\Delta}_j^*$, we write using (12):

$$\log \det (\tilde{\mathbf{A}} + \alpha \boldsymbol{\Delta}_j) = \log \det (\tilde{\mathbf{A}}_{22}) + \log \det(\mathbf{B}_0 + \alpha \mathbf{B}_1 + \alpha^2 \mathbf{B}_2) \tag{14}$$

where $\mathbf{B}_0 = \tilde{\mathbf{A}}_{11} - \tilde{\mathbf{A}}_{12}\tilde{\mathbf{A}}_{22}^{-1}\tilde{\mathbf{A}}_{21}$, $\mathbf{B}_1 = \boldsymbol{\Delta}_{11} - \boldsymbol{\Delta}_{12}\tilde{\mathbf{A}}_{22}^{-1}\tilde{\mathbf{A}}_{21} - \tilde{\mathbf{A}}_{12}\tilde{\mathbf{A}}_{22}^{-1}\boldsymbol{\Delta}_{21}$, and $\mathbf{B}_2 = -\boldsymbol{\Delta}_{12}\tilde{\mathbf{A}}_{22}^{-1}\boldsymbol{\Delta}_{21}$. (Note that here we replaced $\boldsymbol{\Delta}_j^*$ by $\boldsymbol{\Delta}$ to ease notation.)

Finally, the positive definiteness condition $\tilde{\mathbf{A}} + \alpha^* \boldsymbol{\Delta}_j^* \succ 0$ involved in the linesearch (6) is equivalent to $\mathbf{B}_0 + \alpha \mathbf{B}_1 + \alpha^2 \mathbf{B}_2 \succ 0$, assuming that $\tilde{\mathbf{A}}_{22} \succ 0$, following (13). Throughout the iterations, we always guarantee that our iterated solution matrix $\tilde{\mathbf{A}}$ remains positive definite by linesearch in every update. This requires that the initialization of the algorithm, $\mathbf{A}^{(0)}$, be positive definite. If the set $I_j$ is relatively small, then the matrices $\mathbf{B}_i$ in (14) are also small ($|I_j| \times |I_j|$), and we can easily compute the objective $F(\cdot)$, and apply the Armijo rule (6) for $\boldsymbol{\Delta}_j^*$. Calculating the matrices $\mathbf{B}_i$ in (14) seems expensive, however, as we show in Appendix A.3, they can be obtained from the previously computed matrices $W_{I_j}$ and $W_{N_j}$ mentioned earlier. Therefore, computing (14) can be achieved in $O(|I_j|^3)$ time complexity.

---

**Algorithm: BCD-IC($\mathbf{A}^{(0)}$,$\{\mathbf{x}_i\}_{i=1}^m$,$\lambda$)**
**for** $k = 0, 1, 2, ...$ **do**
    Calculate clusters of elements $\{I_j\}_{j=1}^p$ based on $\mathbf{A}^{(k)}$.
    *% Denote:* $\mathbf{A}_0^{(k)} = \mathbf{A}^{(k)}$
    **for** $j = 1, ..., p$ **do**
        Compute $W_{I_j} = \left( (\mathbf{A}_{j-1}^{(k)})^{-1} \right)_{I_j}$. *% solve $|I_j|$ linear systems*
        Define $\mathbf{Active}_{I_j}\left( \mathbf{A}_{j-1}^{(k)} \right)$ as in (9), and define the set $N_j$ in (10).
        Compute $W_{N_j} = \left( (\mathbf{A}_{j-1}^{(k)})^{-1} \right)_{N_j}$. *% solve $|N_j|$ linear systems*
        Find the Newton direction $\boldsymbol{\Delta}_j^*$ by solving the LASSO problem (8).
        Update the solution: $\mathbf{A}_j^{(k)} = \mathbf{A}_{j-1}^{(k)} + \alpha^* \boldsymbol{\Delta}_j^*$ by linesearch.
    **end**
    *% Denote:* $\mathbf{A}^{(k+1)} = \mathbf{A}_p^{(k)}$
**end**

**Algorithm 1:** *Block Coordinate Descent for Inverse Covariance Estimation*

---

## 3 Convergence Analysis

In this Section, we elaborate on the convergence of the BCD-IC algorithm to the global optimum of (3). We base our analysis on [20, 12]. In [20], a general block-coordinate-descent approach is analyzed to solve minimization problems of the form $F(\mathbf{A}) = f(\mathbf{A}) + \lambda h(\mathbf{A})$ composed of the sum of a smooth function $f(\cdot)$ and a separable convex function $h(\cdot)$, which in our case are $-\log \det(\mathbf{A}) + \text{tr}(\mathbf{SA})$ and $\|\mathbf{A}\|_1$, respectively. Although this setup fits the functional $F(\mathbf{A})$ in (3), [20] treats the problem in the $\mathbb{R}^{\mathbf{n}\times\mathbf{n}}$ domain, while the minimization in (3) is being constrained over $\mathbb{S}_{++}^{\mathbf{n}}$—the symmetric positive definite matrices domain. To overcome this limitation, the authors in [12] extended the analysis in [20] to treat the specific constrained problem (3).

In particular, [20, 12] consider block-coordinate-descent methods where in each step $t$ a subset $J_t$ of variables is updated. Then, a Gauss-Seidel condition is necessary to ensure that all variables are updated every $T$ steps:

$$\bigcup_{l=0,...,T-1} J_{l+t} \supseteq \mathcal{N} \quad \forall t = 1, 2, \ldots, \tag{15}$$

where $\mathcal{N}$ is the set of all variables, and $T$ is a fixed number. Similarly to [12], treating each block of columns $I_j$ in the BCD-IC algorithm is equivalent to updating the elements outside the active set $\mathbf{Active}_{I_j}(\mathbf{A})$, followed by an update of the elements in $\mathbf{Active}_{I_j}(\mathbf{A})$. Therefore, in (15), we set

$$J_{2t} = \{(i,l) : i \in I_j \vee l \in I_j\} \setminus \mathbf{Active}_{I_j}(\tilde{\mathbf{A}}), \quad J_{2t+1} = \mathbf{Active}_{I_j}(\tilde{\mathbf{A}}),$$

where the step index $t$ corresponds to the block $j$ at the iteration $k$ of BCD-IC. In [12, Lemma 1], it is shown that setting the elements outside the active set for block $j$ to zero satisfies the optimality condition of that step. Therefore, in our algorithm we only need to update the elements in $\mathbf{Active}_{I_j}(\mathbf{A})$. Now, if we were using $p$ fixed blocks containing all the coordinates of $\mathbf{A}$ in Algorithm (1) (no clustering is applied), then the Gauss-Seidel condition (15) would be satisfied every $T = 2p$ blocks. When clustering is applied, the block-partitioning $\{I_j\}$ can change at every activation of the clustering method. Therefore, condition (15) is satisfied at most after $T = 4\tilde{p}$, where $\tilde{p}$ is the maximum number of blocks obtained from all the activations of the clustering algorithm. For completeness, we include in Appendix A.5 the lemmas in [12] and the proof of the following theorem:

**Theorem 1.** *In Algorithm 1, the sequence* $\left\{\mathbf{A}_j^{(k)}\right\}$ *converges to the global optimum of* (3).

## 4   Numerical Results

In this section we demonstrate the efficiency of the BCD-IC method, and compare it with other methods for both small and large scales. For small-scale problems we include QUIC [12], BIG-QUIC [11] and G-ISTA [8], which are the state-of-the-art methods at this scale. For large-scale problems, we compare our method only with BIG-QUIC as it is the only feasible method known to us at this scale. For all methods, we use the original code which was provided by the authors— all implemented in C and parallelized (except QUIC which is partially parallelized). Our code for BCD-IC is MATLAB based with several routines in C. All the experiments were run on a machine with 2 Intel Xeon E-2650 2.0GHz processors with 16 cores and 64GB RAM, using Windows 7 OS.

As a stopping criterion for BCD-IC, we use the rule as in [11]: $\|grad^S F(\mathbf{A}^{(k)})\|_1 < \epsilon \|\mathbf{A}^{(k)}\|_1$, where $grad^S F(\cdot)$ is the minimal norm subgradient, defined in Equation (25) in Appendix A.5. For $\epsilon = 10^{-2}$ as we choose, this results in the entries in $\mathbf{A}^{(k)}$ being about two digits accurate compared to the true solution $\mathbf{\Sigma}^{-1*}$. As in [11], we approximate $\mathbf{W}_{I_j}$ and $\mathbf{W}_{N_j}$ by using CG, which we stop once the residual drops below $10^{-5}$ and $10^{-4}$, respectively. For stopping NLCG (Algorithm 2) we use $\epsilon_{nlcg} = 10^{-4}$ (see details at the end of Section A.1). We note that for the large-scale test problems, BCD-IC with optimal block size requires less memory than BIG-QUIC.

### 4.1   Synthetic Experiments

We use two synthetic experiments to compare the performance of the methods. First, the `random` matrix from [14], which is generated to have about 10 non-zeros per row, and to be well-conditioned. We generate matrices of sizes $n$ varying from 5,000 to 160,000, and generate 200 samples for each ($m = 200$). The values of $\lambda$ are chosen so that the solution $\mathbf{\Sigma}^{-1*}$ has approximately $10n$ non-zeros. BCD-IC is run with block sizes of 64, 96, 128, 256, and 256 for each of the `random` tests in Table 1, respectively. The second problem is a 2D version of the chain example in [14], which can be represented as the 2D stencil $\frac{1}{4}\begin{bmatrix} & -1 & \\ -1 & 5 & -1 \\ & -1 & \end{bmatrix}$, applied on a square lattice. $\lambda$ is chosen such that $\mathbf{\Sigma}^{-1*}$ has about $5n$ non-zeros. For these tests, BCD-IC is run with block size of 1024.

Table 1 summarizes the results for this test case. The results show that for small-scale problems, G-ISTA is the fastest method and BCD-IC is just behind it. However, from size 20,000 and higher, BCD-IC is the fastest. We could not run QUIC and G-ISTA on problems larger than 20,000 because of memory limitations. The time gap between G-ISTA and both BCD-IC and BIG-QUIC in small-scales can be reduced if their programs receive the matrix $\mathbf{S}$ as input instead of the $\{\mathbf{x}_i\}_{i=1}^m$.

### 4.2   Gene Expression Analysis Experiments

For the large-scale real-world experiments, we use gene expression datasets that are available at the Gene Expression Omnibus (`http://www.ncbi.nlm.nih.gov/geo/`). We use several of the

| test, $\mathbf{n}$ | $\|\Sigma^{-1}\|_0$ | $\lambda$ | $\|\Sigma^{-1*}\|_0$ | **BCD-IC** | **BIG-QUIC** | **QUIC** | **G-ISTA** |
|---|---|---|---|---|---|---|---|
| random 5K | 59,138 | 0.22 | 63,164 | 15.3s(3) | 19.6s(5) | 28.7s(5) | **13.6s(7)** |
| random 10K | 118,794 | 0.23 | 139,708 | 61.8s(3) | 73.8s(5) | 114s(5) | **60.2s(7)** |
| random 20K | 237,898 | 0.24 | 311,932 | **265s(3)** | 673s(5) | 823s(5) | 491s(8) |
| random 40K | 475,406 | 0.26 | 423,696 | **729s(4)** | 2,671s(5) | * | * |
| random 80K | 950,950 | 0.27 | 891,268 | **4,102s(4)** | 16,764s(5) | * | * |
| random 160K | 1,901,404 | 0.28 | 1,852,198 | **21,296s(4)** | 25,584s(4) | * | * |
| 2D $500^2$ | 1,248,000 | 0.30 | 1,553,698 | **24,235s(4)** | 40,530s(4) | * | * |
| 2D $708^2$ | 2,503,488 | 0.31 | 3,002,338 | **130,636s(4)** | 203,370s(4) | * | * |
| 2D $1000^2$ | 4,996,000 | 0.32 | 5,684,306 | **777,947s(4)** | 1,220,213s(4) | * | * |

Table 1: Results for the random and 2D synthetic experiments. $\|\Sigma^{-1}\|_0$ and $\|\Sigma^{-1*}\|_0$ denote the number of non-zeros in the true and estimated inverse covariance matrices, respectively. For each run, timings are reported in seconds and number of iterations in parentheses. '*' means that the algorithm ran out of memory.

tests reported in [9]. The data is preprocessed to have zero mean and unit variance for each variable (i.e., $\mathrm{diag}(\mathbf{S}) = \mathbf{I}$). Table 2 shows the datasets as well as the numbers of variables ($n$) and samples ($m$) on each. In particular, these datasets have many variables and very few samples ($m \ll n$). Because of the size of the problems, we ran only BCD-IC and BIG-QUIC for these test cases.

For the first three tests in Table 2, $\lambda$ was chosen so that the solution matrix has about $10n$ non-zeros. For the fourth test, we choose a relatively high $\lambda = 0.9$ since the low number of samples causes the solutions with smaller $\lambda$'s to be quite dense. BCD-IC is run with block size of 256 for all the tests in Table 2. We found these datasets to be more challenging than the synthetic experiments above. Still, both algorithms BCD-IC and BIG-QUIC manage to estimate the inverse covariance matrix in reasonable time. As in the synthetic case, BCD-IC outperforms BIG-QUIC in all test cases. BCD-IC requires a smaller number of iterations to converge, which translates into shorter timings. Moreover, the average time of each BCD-IC iteration is faster than that of BIG-QUIC.

| code name | Description | $n$ | $m$ | $\lambda$ | $\|\Sigma^{-1*}\|_0$ | **BCD-IC** | **BIG-QUIC** |
|---|---|---|---|---|---|---|---|
| GSE1898 | Liver cancer | $21,794$ | 182 | 0.7 | 293,845 | **788.3s (7)** | 5,079.5s (12) |
| GSE20194 | Breast cancer | $22,283$ | 278 | 0.7 | 197,953 | **452.9s (8)** | 2,810.6s (10) |
| GSE17951 | Prostate cancer | $54,675$ | 154 | 0.78 | 558,929 | **1,621.9s (6)** | 8,229.7s (9) |
| GSE14322 | Liver cancer | $104,702$ | 76 | 0.9 | 4,973,476 | **55,314.8s (9)** | 127,199s (14) |

Table 2: Gene expression results. $\|\Sigma^{-1*}\|_0$ denotes the number of non-zeros in the estimated covariance matrix. For each run, timings are reported in seconds and number of iterations in parentheses.

## 5 Conclusions

In this work we introduced a Block-Coordinate Descent method for solving the sparse inverse co-variance problem. Our method has a relatively low memory footprint, and therefore it is especially attractive for solving large-scale instances of the problem. It solves the problem by iterating and up-dating the matrix block by block, where each block is chosen as a subset of columns and respective rows. For each block sub-problem, a proximal Newton method is applied, requiring a solution of a LASSO problem to find the descent direction. Because the update is limited to a subset of columns and rows, we are able to store the gradient and Hessian for each block, and enjoy an efficient line-search procedure. Numerical results show that for medium-to-large scale experiments our algorithm is faster than the state-of-the-art methods, especially when the problem is relatively hard.

**Acknowledgement:** The authors would like to thank Prof. Irad Yavneh for his valuable comments and guidance throughout this work. The research leading to these results has received funding from the European Union's - Seventh Framework Programme (FP7/2007-2013) under grant agreement no 623212 MC Multiscale Inversion.

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
