[Supplementary Material]

# A Appendix

## A.1 Non-linear Conjugate Gradients for the LASSO problem

The Polak-Ribiere non-linear Conjugate Gradient method was adapted to solve the LASSO problem in [22, 19], and was found in [19] to be the most effective method that is matrix-vector based, and hence, easily parallelizable. Consider the LASSO problem, given in a generic form

$$\min_{\mathbf{x} \in \mathbb{R}^n} G(\mathbf{x}) = \min_{\mathbf{x} \in \mathbb{R}^n} \frac{1}{2}\mathbf{x}^T \mathbf{M}\mathbf{x} - \mathbf{b}^T \mathbf{x} + \lambda \|\mathbf{x}\|_1, \tag{16}$$

where $\mathbf{M} \in \mathbb{R}^{n \times n}$ is a symmetric positive definite matrix and $\mathbf{b} \in \mathbb{R}^n$ is a given vector. This problem corresponds to the LASSO problems in (5) and (8), each one with its corresponding Hessian as $\mathbf{M}$ and gradient as $\mathbf{b}$.

In [22], it was suggested to use a diagonal preconditioner with the non-linear CG method:

$$\mathbf{D} = \text{diag}(\mathbf{M}),$$

i.e., $\mathbf{D}$ is the diagonal part of $\mathbf{M}$. This diagonal preconditioner comes from the shrinkage weights that are used in the coordinate descent method for solving (16). This method uses the so-called soft shrinkage function to overcome the $\ell_1$ term in (16), which is given by

$$\mathcal{S}_\lambda(x) = \text{sign}(x) \cdot \max(0, |x| - \lambda). \tag{17}$$

The full NLCG method as used in [19] is given in Algorithm 2. As in [19], we also use the exact linesearch procedure that appears in [21] as a part of this algorithm.

---

**Algorithm: NLCG($\mathbf{x}^{(0)}$,M,b,$\lambda$)**
Define $\mathbf{z}^{(0)} = 0$.
**for** $k = 1, 2, ...$ **do**
    Define the direction $\mathbf{z}^{(k)} = \mathcal{S}_{\lambda \mathbf{D}^{-1}} \left( \mathbf{x}^{(k)} + \mathbf{D}^{-1}(\mathbf{b} - \mathbf{M}\mathbf{x}^{(k)}) \right) - \mathbf{x}^{(k)}$.
    **if** $nnz(\mathbf{z}^{(k)}) \leq nnz(\mathbf{z}^{(k-1)})$ **then**
        $\beta^{PR} = \max \left\{ 0, \frac{(\mathbf{z}^{(k)})^T(\mathbf{z}^{(k)} - \mathbf{z}^{(k-1)})}{(\mathbf{z}^{(k-1)})^T \mathbf{z}^{(k-1)}} \right\}$.
    **else**
        $\beta^{PR} = 0$.
    **end**
    Update the direction: $\mathbf{z}^{(k)} \leftarrow \mathbf{z}^{(k)} + \beta^{PR}\mathbf{z}^{(k-1)}$.
    Perform a line search: $\alpha^* = \arg\min_\alpha G(\mathbf{x}^{(k)} + \alpha\mathbf{z}^{(k)})$
    Update the solution: $\mathbf{x}^{(k+1)} = \mathbf{x}^{(k)} + \alpha^*\mathbf{z}^{(k)}$
**end**

---

**Algorithm 2:** *Non-linear Conjugate Gradient for LASSO*

The most expensive operation when applying Algorithm 2 to problem (8), is the computation of $\mathbf{M}\mathbf{x}^{(\mathbf{k})}$ (or $\mathbf{M}\mathbf{z}^{(\mathbf{k})}$), which is given by $\mathbf{W}\mathbf{\Delta}_j\mathbf{W}$ in the context of (8).

The preconditioner that we use for NLCG is diagonal, i.e., has a value for each entry of $\mathbf{\Delta}_j$. As said before, the preconditioner corresponds to the shrinkage weights in the coordinate descent method [22] and, hence, following [12] we define the preconditioning weight $\mathbf{\Omega}_{il}$ for the entry $(i, l)$ by

$$\mathbf{\Omega}_{il} = \begin{cases} \mathbf{W}_{ii}\mathbf{W}_{ll} + \mathbf{W}_{il}^2 & i \neq l \\ \mathbf{W}_{ii}^2 & i = l \end{cases}. \tag{18}$$

This is calculated only for the active set (9), and all the values in (18) are known from the computations of $\mathbf{W}_{I_j}$ and $\mathbf{W}_{N_j}$. In this work, similarly to [19], we terminate the NLCG iterations once the correction at iteration $k$ drops below the correction at iteration 1, i.e. when $\|\mathbf{z}^{(k)}\|_1 / \|\mathbf{z}^{(1)}\|_1 < \epsilon_{nlcg}$. Other stopping criterions may be used instead.

## A.2 Dealing with Dense Columns/Rows

In section 2.1.1, we describe how we multiply the Hessian with a direction, i.e., compute $\mathbf{W}(\boldsymbol{\Delta}_j \mathbf{W}_{I_j})$. If the set $N_j$ in (10) is small enough, then $\mathbf{W}_{N_j}$ can be computed and stored in memory. However, computing $\mathbf{W}_{N_j}$ might cause a memory problem if the matrix $\mathbf{A}$ has a few dense columns, and at least one of the sets $N_j$ is of size $O(n)$. We treat this case differently. First, we cluster all the dense columns into one block; let it be $I_1$. Then, we offer two options: one is to re-solve the corresponding $|I_1|$ linear systems for $\mathbf{W}(\boldsymbol{\Delta}_1 \mathbf{W}_{I_1})$ in each time they are needed, that is at each multiplication of the Hessian with a direction in the NLCG algorithm. Since we expect $I_1$ to contain very few columns, this computation should not be too expensive. The other solution is not to process the rows that correspond to the neighborhood $N_1$ for this block, and only focus on the corresponding $|I_1| \times |I_1|$ block of $\mathbf{A}$. Since the entries in the rows of all $N_j$'s are treated twice in each sweep because of symmetry, ignoring them in one block will still result in a convergent method since the union of all blocks still covers all the variables as in Equation (15). Thus, the proof of Theorem 1 still holds in this situations. We note that we did not encounter such a situation in our numerical experiments.

## A.3 Computing the Linesearch Matrices

In this section we describe how to calculate the matrices $\mathbf{B}_i$ in (14) efficiently, using the matrices $W_{I_j}$ and $W_{N_j}$ that are computed before the linesearch procedure (See Algorithm 1). Although the computation of these matrices seems to need the solutions of large linear systems (involving $\mathbf{A}_{22}$ as in (11)), it can be done very efficiently using properties of the Schur complement avoiding the computational burden of solving linear systems.

First, the computation of $\mathbf{B}_0$ is readily available by inverting a small $|I_j| \times |I_j|$ matrix. Since the indices partitioned as '1' are actually those in $I_j$, the matrix $((\mathbf{A}^{-1})_{11})^{-1}$ is available as part of $W_{I_j}$ and by the Schur complement properties we have:

$$\mathbf{B}_0 = ((\mathbf{A}^{-1})_{11})^{-1}. \tag{19}$$

Second, the computation of $\mathbf{B}_1$ is also available by

$$\mathbf{B}_1 = \boldsymbol{\Delta}_{11} + \mathbf{T} + \mathbf{T}^T,$$

where $\boldsymbol{\Delta}$ denotes $\boldsymbol{\Delta}_j$ and

$$\mathbf{T} = -\boldsymbol{\Delta}_{21}^T \mathbf{A}_{22}^{-1} \mathbf{A}_{21} = \boldsymbol{\Delta}_{12}(\mathbf{A}^{-1})_{21}\mathbf{B}_0. \tag{20}$$

The latter is available for us since $(\mathbf{A}^{-1})_{21}$ is again a part of $W_{I_j}$.

For $\mathbf{B}_2 = -\boldsymbol{\Delta}_{12}\mathbf{A}_{22}^{-1}\boldsymbol{\Delta}_{21}$, we need only $\mathbf{A}_{22}^{-1}$ in the block that correspond to $N_j$. That is because $\boldsymbol{\Delta}_{21}$, for example, is non-zero only in the rows that correspond to $N_j$. Following Schur complement we have

$$\mathbf{A}_{22}^{-1} = (\mathbf{A}^{-1})_{22} - (\mathbf{A}^{-1})_{21}\mathbf{B}_0(\mathbf{A}^{-1})_{21}^T, \tag{21}$$

and we need the values of this matrix only at the block that correspond to the columns and rows in $N_j$ (an $N_j \times N_j$ matrix). These, again are available for us from the computation of $W_{N_j}$. Given this matrix, we compute

$$\mathbf{B}_2 = \boldsymbol{\Delta}_{12}^T(\mathbf{A}^{-1})_{22}\boldsymbol{\Delta}_{21} + \left[\boldsymbol{\Delta}_{12}^T(\mathbf{A}^{-1})_{21}\right]\mathbf{B}_0\left[(\mathbf{A}^{-1})_{21}^T\boldsymbol{\Delta}_{21}\right], \tag{22}$$

where the matrix $\boldsymbol{\Delta}_{12}^T(\mathbf{A}^{-1})_{21}$ in brackets is computed also for (20). See next section for more details on the complexity costs of the procedures.

## A.4 Computational complexity and cost

In this section we elaborate on the computational cost of our algorithm. Each BCD-IC iteration consists of treating $p$ blocks, each containing approximately $n/p$ rows/columns. The treatment of

each block has three main stages: (1) computing its gradient, active set and $\mathbf{W}_{N_j}$ for the Hessian. (2) Solving the associated LASSO problem. (3) Applying linesearch. We show that the cost of each BCD-IC iteration depends on the dimension $n$, number of blocks $p$ and the average number of non-zeros per row in $\mathbf{A}^{(k)}$, which we denote by $s$, i.e. $s = \frac{nnz(\mathbf{A}^{(k)})}{n}$. Hence, for example, in the columns of a block $I_j$ there are about $\frac{sn^2}{p}$ non-zeros.

*Computation of the gradient, active set and Hessian.*
This stage is the most expensive part of our algorithm, and it is dominated by the computation of $\mathbf{S}_{I_j}$, $\mathbf{W}_{I_j}$, $\mathbf{W}_{N_j}$. The first two are required for the gradient and active set (9), while the last is required for the Hessian. Computing $\mathbf{S}_{I_j}$ is done in $mn|I_j|$ computations for each block $I_j$ ($m$ is the number of samples $\mathbf{x}_i$), and overall for all blocks it sums to $mn^2$.

As explained before, each of the columns of $\mathbf{W}_{I_j}$ and $\mathbf{W}_{N_j}$ is computed by solving a linear system $\mathbf{Az} = \mathbf{e}_l$ where $\mathbf{e}_l$ is the suitable canonical vector. In this work, we use CG to solve these linear systems, and assume that each system solution requires $T_{cg}$ iterations, each one dominated by a matrix-vector multiplication. Hence, solving the $|I_j| + |N_j|$ linear systems requires $O(T_{cg}(|I_j| + |N_j|)sn)$ operations.

Overall, computing the gradient, the active set in (9) and $\mathbf{W}_{N_j}$ for the Hessian for all blocks sums to $O\left(mn^2 + T_{cg}sn^2 + T_{cg}sn\sum_{j=1}^{p}|N_j|\right)$. While the first two terms do not depend on the number of blocks $p$, the last one decreases as $p$ grows. For $p = 1$ (no partitioning) $|N_j| = 0$, and for $p = n$ $\sum_{j=1}^{p}|N_j| = sn$. For an arbitrary $p$, $\sum_{j=1}^{p}|N_j|$ depends on the connectivity of the graph of $A$.

*Solution of the LASSO Problem.*
Similarly to the CG algorithm for solving linear systems, the cost of the non-linear CG method is dominated by the number of matrix-vector multiplications (one at each iteration). Again, we assume that each solve of the block sub-problem (8) requires $T_{nlcg}$ iterations. Following Section (2.1.1), each Hessian computation $(\mathbf{W}\mathbf{\Delta}_j\mathbf{W})_{I_j}$ restricted to entries in the active set (9) requires $O(s|I_j|(|N_j|+|I_j|))$ operations. Overall, for a whole sweep of BCD-IC, the NLCG method requires $O\left(snT_{nlcg}(\frac{n}{p} + \sum_{j=1}^{p}|N_j|)\right)$. This cost has two terms with opposite dependence on $p$: as $p$ grows the term $\sum_{j=1}^{p}|N_j|$ also grows, but the term $\frac{n}{p}$ reduces.

In the method [11], a similar block-coordinate-descent approach is applied on the full LASSO problem (5), instead of the original problem (3) as in our case. There, each sweep costs roughly $O(n^2)$ operations, versus $O(\frac{n^2}{p})$ per full sweep as in our case. That is another advantage of our algorithm compared to [11], since we assume that $p$ is large for large-scale problems.

*The linesearch procedure.*
Following the description in Sections 2.1.2 and A.3, the linesearch procedure requires several dominating computations. First, Equations (20) and (22) involve three multiplications of a sparse matrix with a dense matrix, requiring $O(s|I_j|^2) + O(s|I_j||N_j|)$ operations. More dominant, however, are the dense matrix multiplications in Equations (19), (20), (22), and computing the determinants according to (14) in the Armijo linesearch procedure (6). Each of these cost $O(|I_j|^3)$ operations, often dominating the sparse-dense multiplications. Overall, the linesearch procedure for all the $p$ blocks is achieved in $O(n\frac{n^2}{p^2}) + O\left(sn(\frac{n}{p} + \sum_{i=1}^{p}|N_j|)\right)$. Here, again we have terms with opposite dependence on $p$.

*The overall cost of the BCD-IC iteration.*
Summing all the complexity costs above, we conclude the the overall cost of the whole algorithm is dominated by

$$O\left(mn^2 + sn^2\left[T_{cg} + \frac{T_{nlcg}}{p} + (T_{cg} + T_{nlcg})\frac{1}{n}\sum_{j=1}^{p}|N_j|\right] + \frac{n^3}{p^2}\right)$$

operations. The only two quantities that do not depend on $p$ are the first ones above, needed for computing the gradient and active set. The rest of the costs either grow as $p$ grows, or reduce as $p$ grows. This suggests that there is an optimal value for $p$ to reduce the cost of the entire algo-

rithm. This value depends on the unknown parameters of the problem—the condition numbers of the matrices which influences $T_{cg}$, the connectivity of the graph which influences $N_j$, the sparsity $s$, etc. In addition, the optimal value of $p$ depends on the machine characteristics where the algorithm runs. For example, the block size should be several times the number of cores in the machine in order to obtain good parallelization performance. Figure 2 shows the runtime for several choices of block size for the experiment with the random matrix (size 10,000) which appears in Table 1. The presented timings are averaged over 3 experiments, and it is clear that there is an optimal choice for the block size.

Figure 2: Solution timings per block size for a random problem, $n = 10,000$.

From all of the above problem parameters, the one which is hardest to predict is $\sum_{j=1}^{p} |N_j|$, and is the only term that forces us to choose fewer blocks (choose $p$ to be small). Figure 3 shows the sum of neighborhoods per number of blocks for two problems. One is the random problem in Table 1 and the other involves a matrix that corresponds to a simple 2D cartesian lattice with five non-zeros per row. It shows that $\sum_{j=1}^{p} |N_j|$ is relatively high for the random case, which forces us to choose a relatively small number of blocks. On the other hand, as we increase the number of blocks for the other example, the sum of neighborhoods remains relatively small, so we expect to choose a high number of blocks for this problem and reduce the costs of the other stages of the algorithm.

Figure 3: Sum of neighborhoods $\sum_{j=1}^{p} |N_j|$ per number of blocks $p$.

## A.5   Convergence Guarantee

In this section we first recall the lemmas in [12], and then we use them to prove Theorem 1. For that purpose, we introduce some useful notation. First, we denote by $\mathbf{A}$ the estimated matrix before the treatment of a block. We assume that $\mathbf{\Delta}_j$ is the Newton's direction $\mathbf{\Delta}_j^*(\mathbf{A})$ obtained for solving (8) for matrix $\mathbf{A}$ and some block $j$ at some iteration of Algorithm 1. Furthermore, we treat the matrices such as $\mathbf{\Delta}_j$ by their actual vectorized version $\text{vec}(\mathbf{\Delta}_j)$, such that for example, $\mathbf{\Delta}_j \nabla^2 f(\mathbf{A}) \mathbf{\Delta}_j$ appears in the equations instead of $\text{vec}(\mathbf{\Delta}_j) \nabla^2 f(\mathbf{A}) \text{vec}(\mathbf{\Delta}_j)$. Next, we denote by $\delta_j(\mathbf{A})$ the term in Armijo's rule (6),

$$\delta_j(\mathbf{A}) = \text{tr}(\mathbf{\Delta}_j(\mathbf{S} - \mathbf{W})) + \lambda \|\mathbf{A} + \mathbf{\Delta}_j\|_1 - \lambda \|\mathbf{A}\|_1. \tag{23}$$

### A.5.1 Lemmas

The following Lemmas are related to the specific problem (3) that we are solving. Lemma 1 suggest that for any algorithm, all the iterates have positive bounded eigenvalues. Lemmas 2 and 3 define optimality and the properties of a stationary point.

**Lemma 1.** *The level set* $U = \left\{ \mathbf{A} | F(\mathbf{A}) < F(\mathbf{A}^{(0)}) \text{ and } \mathbf{A} \in \mathbb{S}^{\mathbf{n}}_{++} \right\}$ *is contained in the set* $\{\mathbf{A} | m\mathbf{I} \preceq \mathbf{A} \preceq M\mathbf{I}\}$ *for positive constants* $m, M > 0$.

**Lemma 2.** *Problem* (3) *has a unique global minimizer* $\mathbf{A}^*$.

**Lemma 3.** $\mathbf{A}^*$ *is the optimal solution of* (3) *if and only if*

$$grad_{ij}^S F(\mathbf{A}^*) = 0 \quad \forall i, j, \tag{24}$$

*where the minimum-norm sub-gradient* $grad_{ij}^S F(\mathbf{A})$ *is defined by*

$$grad_{ij}^S F(\mathbf{A}) = \begin{cases} \nabla_{ij} f(\mathbf{A}) + \lambda & if \; \mathbf{A}_{ij} > 0, \\ \nabla_{ij} f(\mathbf{A}) - \lambda & if \; \mathbf{A}_{ij} < 0, \\ \text{sign}(\nabla_{ij} f(\mathbf{A})) \max\left( |\nabla_{ij} f(\mathbf{A})| - \lambda, 0 \right) & if \; \mathbf{A}_{ij} = 0. \end{cases} \tag{25}$$

Next, Lemma 4 introduces a direct connection between the Newton's direction obtained for a block and the sub-gradient value in a stationary point.

**Lemma 4.** *For any index set* $I \subseteq N$, $\boldsymbol{\Delta}_I(\mathbf{A}) = \boldsymbol{\Delta}_I = 0$ *if and only if* $grad_{ij}^S F(\mathbf{A}) = 0$ *for all* $(i, j) \in I$.

The Lemma below proves that the linesearch term (23) is always negative or zero, implying the functional decrease after the lineseach procedure (6).

**Lemma 5.** *The term* $\delta_j(\mathbf{A})$ *in the line search condition* (6) *for* $\boldsymbol{\Delta}_j$ *satisfies*

$$\delta_j(\mathbf{A}) = \nabla f(\mathbf{A})^T \boldsymbol{\Delta}_j + \lambda \|\mathbf{A} + \boldsymbol{\Delta}_j\|_1 - \lambda \|\mathbf{A}\|_1 \le -\boldsymbol{\Delta}_j^T \nabla^2 f(\mathbf{A}) \boldsymbol{\Delta}_j, \tag{26}$$

*and consequently,*

$$\delta_j(\mathbf{A}) \le -m \|\boldsymbol{\Delta}_j\|_F^2. \tag{27}$$

Additionally, it is necessary to show that the linesearch procedure always can find an $\alpha$ value that maintains the positive definiteness of the updated estimate $\mathbf{A} + \alpha\boldsymbol{\Delta}$. This is proven with the following Lemma:

**Lemma 6.** *For any* $\mathbf{A} \succ 0$ *and symmetric* $\boldsymbol{\Delta}$, *there exists an* $\bar{\alpha} > 0$ *such that for all* $\alpha < \bar{\alpha}$, *(1)* $\mathbf{A} + \alpha\boldsymbol{\Delta} \succ 0$ *and (2)* $\mathbf{A} + \alpha\boldsymbol{\Delta}$ *satisfies the line search condition* (6).

The last Lemma proves that any convergent sequence of iterates in the algorithm has the Newton's direction converging to the zero matrix. This Lemma proves implicitly that the functional is nonincreasing between iterates and these converge to a stationary point.

**Lemma 7.** *For any convergent subsequence* $\mathbf{A}_{s_t} \to \bar{\mathbf{A}}$,

$$\boldsymbol{\Delta}_{s_t} \equiv \boldsymbol{\Delta}_{J_{s_t}}(\mathbf{A}_{s_t}) \to 0. \tag{28}$$

### A.5.2 Proof of Theorem 1

Next, we use the above Lemmas to prove Theorem 1 presented in Section 3.

*Proof.* Suppose $\{\mathbf{A}_t\}$ is a convergent sequence to $\bar{\mathbf{A}}$ obtained from our algorithm with the blocks satisfying the Gauss-Seidel condition (15). Let $\{\mathbf{A}_t\}_T$ be a subsequence of $\{\mathbf{A}_t\}$ converging to $\bar{\mathbf{A}}$. Since the choice of the index set $I_t$ selected at each step is finite, we can further assume that $I_t = \bar{I}_0$ for all $t \in T$. From Lemma 7, $\boldsymbol{\Delta}_{\bar{I}_0}(\mathbf{A}_t) \to 0$. By the continuity of $\nabla f(\mathbf{A})$ and $\nabla^2 f(\mathbf{A})$, it follows that $\boldsymbol{\Delta}_{\bar{I}_0}(\mathbf{A}_t) \to \boldsymbol{\Delta}_{\bar{I}_0}(\bar{\mathbf{A}})$. Hence, $\boldsymbol{\Delta}_{\bar{I}_0}(\bar{\mathbf{A}}) = 0$.

Furthermore, $\left\{\mathbf{\Delta}_{\bar{I}_0}(\mathbf{A}_t)\right\}_T \to 0$ and $\|\mathbf{A}_t - \mathbf{A}_{t+1}\|_F \le \|\mathbf{\Delta}_{\bar{I}_0}(\mathbf{A}_t)\|_F$, so $\{\mathbf{A}_{t+1}\}_T$ also converges to $\bar{\mathbf{A}}$. By further subsetting of $T$ we can assume that $I_{t+1} = \bar{I}_1$ for all $t \in T$. By the same argument we can prove $\left\{\mathbf{\Delta}_{I_{t+1}}(\mathbf{A}_t)\right\}_T \to 0$, so $\mathbf{\Delta}_{\bar{I}_1}(\bar{\mathbf{A}}) = 0$. Similarly, we can show that $\mathbf{\Delta}_{\bar{I}_i}(\bar{\mathbf{A}}) = 0 \ \forall i = 0, \ldots, T-1$ can be assumed for an appropriate subset of $T$. According to Lemma 4 and the Gauss-Seidel condition (15), $\bar{\mathbf{A}}$ is a stationary point: $grad_{ij}^S F(\bar{\mathbf{A}}) = 0 \ \forall i, j$. Moreover, by Lemma 2, there exists a unique optimal point, so the sequence $\{\mathbf{A}_t\}$ generated by our algorithm must converge to the global optimum. $\qquad\square$