[Reviews · NeurIPS 2014]

Submitted by Assigned_Reviewer_22

In this paper, the authors presented a new block-coordinate descent approach for the sparse inverse covariance estimation problem. In this method, the sought matrix is treated block-by-block using quadratic approximations.

This paper is clearly written and technically sound. Experiments using both synthetic and real data demonstrate that the proposed method outperforms several other methods in many cases. In this sense, the paper is valuable. However, the novelty of this paper is not clear from the current writing. Various blocking strategies have been proposed for Newton method and for solving LASSO problems. It's worth to do a more comprehensive comparison with other blocking methods, both in experiments and conceptually.
Summary: The paper can be marginally accepted upon revision. While the experiments show favorable results, the novelty of the paper is not clear (i.e. whether the improvements are results of the fine tuning of parameters or results of a major idea).

Submitted by Assigned_Reviewer_27

This paper proposes a more efficient, block decomposition, algorithm for reconstruction of large sparse inverse covariance matrices. The paper is written clearly. The experimental results show considerable gains in speed and memory for comparable reconstruction accuracy. I am not an expert in this area so I cannot speak to the work's originality, but as far as I could understand the details, the method appears sound. The experimental results seem solid but not ground-breaking because the problems are still small enough to run on a single machine. It would have been nice to see an extension to the really large problems (>10^6 variables) that one sees in some applications by exploting distributed computation and sharding of the various relevant matrices. One concern for replicating the work is that there seems to be some "magic" in setting up the block decomposition and in the line search. It would have been nice to have a discussion of the setting of those choices.
Summary: Well-presented improvement to methods for sparse inverse covariance reconstruction. The kind of solid incremental advance that moves practice forward and is likely to be replicated and extended by others.

Submitted by Assigned_Reviewer_44

Summary of the paper. Memory can be an issue when estimating a large-scale inverse covariance matrix. In the setting where an l^1 penalty is used, the authors propose a block approach that significantly reduces memory usage while satisfying convergence guarantees.

General comments.
The paper is very clear and well written, and the problem is of importance. As a non-expert, I cannot really judge of the novelty of the method, but the paper seems to be a combination of references [11, 12, 21] in particular. Experiments are convincing.

Minor comments.
Eqn 15: you call the sets of variables J while two lines earlier they are denoted by I.
Summary: The paper is very clear and well written, and the problem is of importance. As a non-expert, I cannot really judge of the novelty of the method, but the paper seems to be a combination of references [11, 12, 21] in particular.
Author Feedback
Author rebuttal: We thank the reviewers for their positive assessment of our paper and valuable comments.

Reviewer 22:

"The novelty of the paper is not clear (i.e. whether the improvements are results of the fine tuning of parameters or results of a major idea)."

The focus of our paper is on the solution of the sparse inverse covariance estimation problem at large scales. The only existing method that is capable of handling large-scale problems (problems where A^{-1} cannot be stored in memory) is [11]. Our method is significantly different from the existing methods, including [11], and contains many favorable features.

Our algorithm is a novel block-coordinate descent approach for solving the sparse inverse covariance estimation in large scale dimensions, where each block is solved by Newton's method. In contrast, [11] is a Newton approach for the complete problem, where the LASSO problem of computing the sparse n x n Newton direction is solved approximately by coordinate descent in blocks.

We demonstrate that our method is computationally faster than [11] for medium to large-scale problems. Each iteration of our method is faster than an iteration of [11], as a result of the reasons listed below. On top of this, we demonstrate empirically that our method also converges in less iterations than [11].

More specifically, our approach introduces several advantages over the method of [11]:
1. The computation of A^{-1} (each block of it at a time) is usually the most expensive step for solving the sparse inverse covariance problem in large scales. Because our descent directions are computed for each block separately, the respective columns of A^{-1} are computed once per block. They are then used to compute the gradient and the Hessian for solving the LASSO problem (the computation of W_{I_j} in Algorithm 1 in our paper). In contrast, in [11] the descent directions are n x n sparse matrices, and A^{-1} (each block of it at a time) is computed at least twice per block (Steps 2 and 6 in Algorithm 1 in [11,Section 3.3]): once for the active set (and gradient), and once for the Hessian in each iteration of the LASSO iterative solver (Coordinate Descent). This would be necessary in [11] for any method that can be used to solve the LASSO problem.
2. In addition to this, also the columns of A^{-1} that correspond to the neighborhood of each block, W_{N_j}, are computed once per block for each iteration of the LASSO iterative solver, while in our method W_{N_j} is computed only once for each block.
3. Because our descent directions are limited only to each block, our method enjoys a highly efficient line-search procedure. We achieve this by exploiting the mathematical properties of the Schur complement, as is summarized in equations (11)-(14). Hence, in our algorithm this procedure is almost negligible compared to the other parts. In [11], on the other hand, the determinant of an n x n matrix needs to be computed in the line-search procedure, which is a costly procedure.

We note that for small problems our method is comparable to other state-of-the-art methods.

Various blocking strategies have been proposed for Newton method and for solving LASSO problems. It's worth to do a more comprehensive comparison with other blocking methods, both in experiments and conceptually

The papers [8,11,12,16] show that they are the state-of-the-art methods for our problem at small scale, and therefore we compare our method to them. The general block-coordinate-descent approach was analyzed in [21] for a general non-smooth convex optimization. Although we follow [21] for the inverse covariance estimation problem, the problem domains are quite different. Our problem is over the positive definite matrices domain, whereas [21] considers the entire domain n-dimensional vector space (R^n).

We would be happy to include a comparison with more methods in a revision, but we need to better understand the intent of the reviewer.

Reviewer 27:
"It would have been nice to see an extension to the really large problems ( > 10^6 variables) that one sees in some applications by exploting distributed computation and sharding of the various relevant matrices."

We agree. We plan to add results for larger problems in the final revision of the paper. As an example, using a 2D-lattice synthetic experiment with dimension n=500,000, k=200 samples, and lambda = 0.31, BCD-IC took 130,636sec (~36.2 hours) and 4 iterations to finish in the same machine reported in the paper.

"One concern for replicating the work is that there seems to be some "magic" in setting up the block decomposition and in the line search. It would have been nice to have a discussion of the setting of those choices."

The column partitioning is performed according to a graph clustering algorithm, and described after Equation (10). This approach is taken from [11], and it is useful for reducing the cost of the Hessian computations. It is fully defined.

The highly efficient line-search procedure is a direct result of our choice of block decomposition and mathematical properties of the Schur complement. Beyond equations (6) and (11)-(14), a detailed description of the line-search procedure and its computational cost is summarized in the supplementary material. This mathematical description is quite technical, hence included only there. However, it provides a solid mathematical foundation for the efficiency of the line-search. Once again, it is fully defined.

The number of blocks is a significant parameter of the algorithm. A detailed analysis of all the costs of the different phases in the algorithm (as a function of this parameter) appears in the supplementary material in A.4.

We note that all the details of the algorithm are given in the paper, such that readers can replicate the work. In addition, we plan to release our code after the final revision.

Reviewer 44:

Minor comments will be corrected.